# Simulation Study on Gas Holdup of Large and Small Bubbles in a High Pressure Gas–Liquid Bubble Column

**Fangfang Tao, Shanglei Ning, Bo Zhang, Haibo Jin * and Guangxiang He ***

Beijing Key Laboratory of Fuels Cleaning and Advanced Catalytic Emission Reduction Technology, School of Chemical Engineering, Beijing Institute of Petrochemical Technology, Beijing 102617, China

* Correspondence: jinhaibo@bipt.edu.cn (H.J.); hgx@bipt.edu.cn (G.H.); Tel.: +86-010-8129-2208 (H.J.)

**Abstract:** The computational fluid dynamics-population balance model (CFD-PBM) has been presented and used to evaluate the bubble behavior in a large-scale high pressure bubble column with an inner diameter of 300 mm and a height of 6600 mm. In the heterogeneous flow regime, bubbles can be divided into "large bubbles" and "small bubbles" by a critical bubble diameter *dc*. In this study, large and small bubbles were classified according to different slopes in the experiment only by the method of dynamic gas disengagement, the critical bubble diameter was determined to be 7 mm by the experimental results and the simulation values. In addition, the effects of superficial gas velocity, operating pressure, surface tension and viscosity on gas holdup of large and small bubbles in gas–liquid two-phase flow were investigated using a CFD-PBM coupling model. The results show that the gas holdup of small and large bubbles increases rapidly with the increase of superficial gas velocity. With the increase of pressure, the gas holdup of small bubbles increases significantly, and the gas holdup of large bubbles increase slightly. Under the same superficial gas velocity, the gas holdup of large bubbles increases with the decrease of viscosity and the decrease of surface tension, but the gas holdup of small bubbles increases significantly. The simulated values of the coupled model have a good agreement with the experimental values, which can be applied to the parameter estimation of the high pressure bubble column system.

**Keywords:** high pressure bubble column; the critical bubble diameter; the gas holdup; the large bubbles; the small bubbles

## 1. Introduction

As a common multi-phase reactor, the bubble column reactor has been widely used in petrochemical, fermentation, waste water treatment, mineral processing and metallurgy industries due to its lack of mechanically operated parts, large phase-contacting area, easy operation, high mass transfer and heat transfer efficiency [1–3]. The flow structure in the bubble column is greatly influenced by gas–liquid properties, gas flow rate, bubble size and distributor design [4]. Gas holdup is the volume fraction of gas in the total volume of gas–liquid phase in the bubble column, which is one of the most important parameters to characterize the hydrodynamic characteristics of the bubble column. It is closely related to the bubble size and the superficial gas velocity [5]. Moreover, these parameters will be more or less affected by the hydrodynamic characteristics, such as the style of the gas distributor [6], the column diameter, the liquid height [7] and the liquid properties [8]. It is a focus issue for the design, optimization and scale-up of the bubble column reactor to optimize these operation parameters to improve the gas holdup and phase interface area in the column.

From the mesoscopic scale, there are multi-scales of bubbles in the bubble column. In the homogeneous flow regime, the small sized and uniform bubbles are generated by the gas distributor,

and the bubbles rise slowly. However, in the heterogeneous flow regime, because of coalescence and breakup, bubbles can be divided into "large bubbles" and "small bubbles" by a critical bubble diameter $dc$ with a two-bubble-class hydrodynamic model [9–11]. Generally, the small bubbles are in the range of 3–6 mm [10], and the large bubbles are typically in the range of 10–30 mm [10,12]. Therefore, it plays an important role in studying the gas holdup of large and small bubbles, as well as in accurately predicting flow patterns and gas dynamics in the mesoscopic scale [13]. In this study, large and small bubbles were classified only by the different slopes measured by the dynamic gas disengagement method in the experiment [9]. At present, the effects of different factors on gas holdup have been mainly investigated through experimental techniques (dynamic gas disengagement, differential pressure [14], electrical resistance tomography, conductivity probe [15], etc.) and computational fluid simulation. Among them, Besagni et al. [16] used experimental measurements and image analysis to investigate flow regime transition, bubble column hydrodynamics, bubble shapes, and size distributions. Gemello et al. [17] used two optical probes to investigate the effects of contaminants and spargers on the bubble size, and they found that adding contaminants and alcohol in the bubble column inhibited bubble coalescence and caused the decrease of bubble sizes. The data obtained by experimental techniques can provide reliable tools for the validation of the CFD-PBM model. Zhang et al. [18] used dynamic gas disengagement (DGD) to investigate the effects of surface tension, viscosity, pressure and superficial gas velocity on the gas holdup of large and small bubbles. Yang et al. [19] showed that the critical bubble sizes in the acetic acid-air experimental system are 5 and 6 mm by using DGD. The effects of operating pressure (0–2.0 MPa) and superficial gas velocity (0.08–0.32 m/s) on the gas holdup of large and small bubbles were also investigated. The results showed that with the increase of pressure, the gas holdup of small bubbles increased obviously, and with the increase of superficial gas velocity, the gas holdup of large bubbles changed greatly while the gas holdup of small bubbles did not change significantly. Xing [20] also used the DGD method and computational fluid dynamics (CFD) simulations to investigate the influence of superficial gas velocity on the characteristics of bubbles in a deionized water–air system, and they showed a powerful function in different fields for estimating the hydrodynamic behavior in the bubble column, providing a reasonable basis for the design and amplification of the bubble column [21]. CFD is helpful to understand the characteristics of fluid flow by combining simulation results with the experimental results [22], the wake acceleration effect of large and small bubbles, as well as the effect of operating pressure on bubble collapse were considered by Yang et al. [23]. They also used the modified CFD-PBM coupling model to investigate the effect of pressure variation on gas holdup under different superficial gas velocities (0.04–0.16 m/s). The results showed that the gas holdup of small bubbles increased rapidly with the increase of pressure, while the gas holdup of large bubbles basically remained unchanged or decreased slightly.

Most of the previous simulations have focused on changes in total gas holdup under different conditions. However, the distribution characteristics of gas holdup of large and small bubbles are rarely simulated.

The aim of this study was to divide the bubbles into large and small bubbles from the mesoscopic scale, and the coupling CFD-PBM model was used to simulate the results of the literature and present the effect of superficial gas velocity, operating pressure, surface tension, viscosity and other conditions on the gas holdup of large and small bubbles. Through numerical simulation, it was found that the simulation results were basically consistent with the experimental results, and the effect of superficial gas velocity on the gas holdup of large and small bubbles can be well predicted.

## 2. Experimental Setup

The experimental setup is shown in Figure 1. The material of the bubble column was stainless steel to meet the high pressure experimental conditions. The inner diameter of the reactor was 300 mm, and the height was 6600 mm. The conductivity probe, the differential pressure method and the ERT (electrical resistance tomography) were installed in the height range of 2500–3100 mm on both sides of the bubble column to get the gas holdup. The three measuring methods in different plane had a

good reliability in measuring gas holdup, as shown in Figure 2a,b. The experimental system was an air–water system under operating pressure (0.5–2.0 MPa) and superficial velocity (0.16–0.32 m/s). The hydrodynamic parameters in the high pressure bubble column were measured and analyzed by different testing techniques. Among them, two ERT electrode matrices were mounted on two cross-sections of the bubble column height of 3000 (Plane 1) and 2600 mm (Plane 2), and each electrode matrix was composed of 16 electrodes installed in the bubble column.

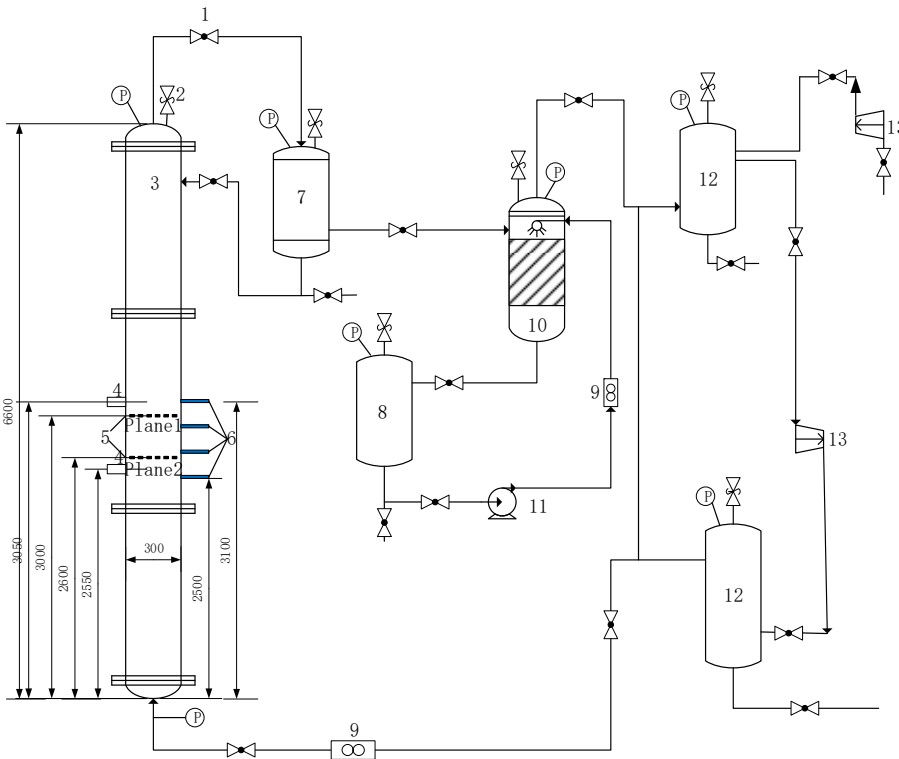

**Figure 1.** Schematic diagram of the experiment. **1**: Valve; **2**: Safe vale; **3**: Bubble column; **4**: Conductivity probe; **5**: Electrodes of electrical resistance tomography (ERT); **6**: Differential pressure measuring pin; **7**: Gas–liquid separation tank; **8**: Liquid storage tank; **9**: Vortex flowmeter; **10**: Absorption column; **11**: Pump; **12**: Gas storage tank; **13**: Air compressor.

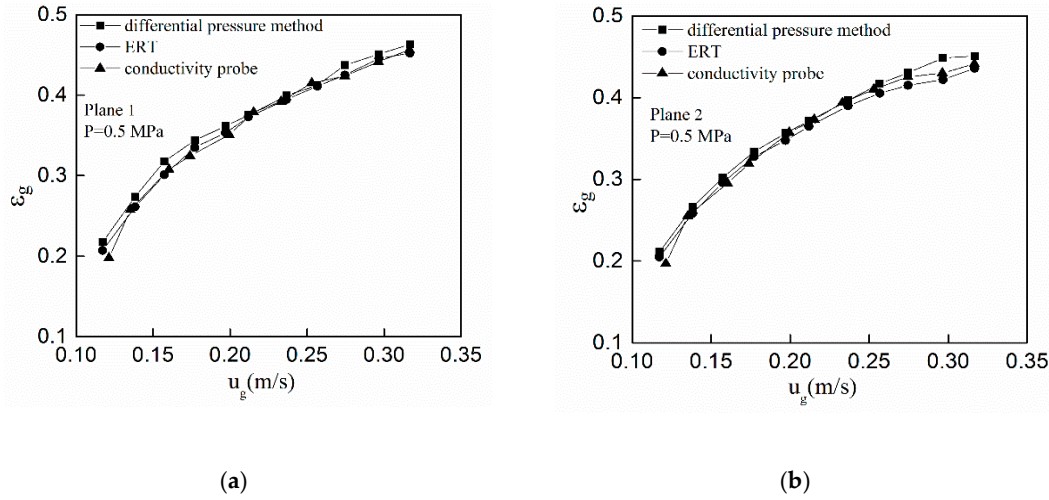

| (a) | (b) |

**Figure 2.** (**a**) The gas holdup $\varepsilon_g$ measured by three methods at plane 1 under different superficial gas velocities; (**b**) The gas holdup $\varepsilon_g$ measured by three methods at plane 2 under different superficial gas velocities.

## 3. Mathematical Model

*3.1. Two-Fluid Model*

In this study, using the FLUENT 15.0 as the platform. In the Euler-Euler model, both water and bubbles were considered as the continuous phase of the calculation zone. The model equation mainly included the continuum equation and the momentum conservation equation. The specific expression is as follows:

Continuum equation: ($i$ = liquid or gas phase)

$$\frac{\partial \alpha_i}{\partial t} + \nabla \cdot (\alpha_i U_i) = 0 \tag{1}$$

Momentum conservation equation: ($i$ = liquid or gas phase)

$$\frac{\partial \alpha_i \rho_i U_i}{\partial t} + \nabla \cdot (\alpha_i \rho_i U_i U_i) = -\alpha \cdot \nabla p + \nabla \cdot (\alpha_i \tau_i) + (-1)^i F_i + \alpha_i \rho_i g \tag{2}$$

*3.2. Interphase Force*

3.2.1. Drag Force

This study was based on the bubble swarm drag model of Roghair et al. [24]. In the heterogeneous flow regime bubbles in the column can coalescence and break up, it can be seen that the bubble size distribution in the column was wide, which was different from the drag force of single bubble size. The drag force of the bubble swarm is not only related to the liquid phase but is also affected by the interaction between the bubbles. The experimental values of Qin [25] under different superficial gas velocities (0.088–0.317 m/s) and pressures (0.1–2.0 MPa) were used to correct a semi-empirical bubble swarm drag force, as shown in Table 1. The density correction term $\rho_g/\rho_0$ was introduced into the bubble swarm model, and the modified bubble group drag force model is shown in Equation (3).

**Table 1.** Bubble group drag coefficient for numerical simulation.

| P(MPa) | $\rho_g/\rho_0$ | $u_g$ | $d_{B,exp}$ | $\varepsilon_{g,exp}$ | $C_D$ | $\varepsilon_{g,sim}$ | Error |
|---|---|---|---|---|---|---|---|
| 0.1 | 1 | 0.088 | 9.64 | 0.17 | 0.70 | 0.18 | 0.17% |
| | | 0.132 | 9.91 | 0.22 | 0.40 | 0.21 | −1.94% |
| | | 0.154 | 10.05 | 0.23 | 0.28 | 0.22 | −2.08% |
| | | 0.199 | 10.32 | 0.25 | 0.21 | 0.25 | 0.51% |
| 0.5 | 5 | 0.199 | 9.67 | 0.35 | 0.51 | 0.36 | 0.42% |
| | | 0.233 | 9.84 | 0.39 | 0.46 | 0.39 | −0.83% |
| | | 0.275 | 10.08 | 0.42 | 0.41 | 0.42 | −0.81% |
| | | 0.317 | 10.30 | 0.45 | 0.36 | 0.44 | −0.97% |
| 1.0 | 10 | 0.199 | 9.39 | 0.35 | 0.51 | 0.36 | 0.42% |
| | | 0.233 | 9.83 | 0.49 | 0.68 | 0.49 | −0.93% |
| | | 0.275 | 10.13 | 0.51 | 0.55 | 0.50 | −1.09% |
| | | 0.317 | 10.22 | 0.52 | 0.46 | 0.53 | −0.78% |
| 1.5 | 15 | 0.199 | 9.15 | 0.50 | 0.88 | 0.51 | 0.50% |
| | | 0.233 | 9.69 | 0.53 | 0.75 | 0.53 | −0.96% |
| | | 0.275 | 9.93 | 0.55 | 0.63 | 0.55 | 0.11% |
| | | 0.317 | 10.06 | 0.56 | 0.49 | 0.56 | 0.82% |
| 2.0 | 20 | 0.199 | 8.98 | 0.54 | 0.97 | 0.54 | 0.37% |
| | | 0.233 | 9.14 | 0.56 | 0.78 | 0.56 | −0.27% |
| | | 0.275 | 9.36 | 0.57 | 0.63 | 0.57 | −0.64% |
| | | 0.317 | 9.50 | 0.58 | 0.52 | 0.58 | −0.41% |

$$\frac{C_D}{C_{D,\infty}(1-\varepsilon_G)} = 3.94\left(\frac{\rho}{\rho_0}\right)^{-0.70} + \left[\frac{97.20}{E_0} - 19.84\left(\frac{\rho}{\rho_0}\right)^{-0.73}\right]\varepsilon_G \tag{3}$$

### 3.2.2. Turbulent Dispersion Force

Generally, the fluid flow in the bubble column is in a turbulent state. In order to describe the turbulent action in the liquid phase, it is necessary to introduce a turbulent dispersion force, because the introduction of a turbulent diffusion force contributes to uniform calculation results of the gas holdup in the column, which makes it more consistent with the experimental values. In this section, we used the turbulent dispersion force proposed by Lopez de Bertodano [26]. The specific expressions are listed as follows:

$$F_{TD,L} = -F_{TD,G} = C_{TD}\rho_L k_L \nabla \varepsilon_G \tag{4}$$

where $C_{TD}$ is the turbulent diffusion force coefficient, and its default value is 1.

It is difficult to converge when the numerical simulation is carried out directly using Equation (4). In the FLUENT 15.0 platform, the limit function $f_{TD,limiting}$ is added to the model. Therefore, the modified model of the turbulent dispersion force is given as follows:

$$F_{TD,L} = -F_{TD,G} = f_{TD,limiting}C_{TD}\rho_L k_L \nabla \varepsilon_G \tag{5}$$

$$f_{TD,limiting}(\varepsilon_G) = \max\left(0, \min\left(1, \frac{\varepsilon_{G,2} - \varepsilon_G}{\varepsilon_{G,2} - \varepsilon_{G,1}}\right)\right) \tag{6}$$

where $\varepsilon_{G,1}$ is 0.3 and $\varepsilon_{G,2}$ is 0.7.

### 3.2.3. Horizontal Lift Force

When the bubble moves upward in the column, the pressure distribution around the bubble is unbalanced due to the asymmetry of the liquid in the direction of movement of the bubble [27]. This produces a horizontal lift force perpendicular to the direction of motion of the bubble. Drew [28] proposed that the lift force experienced in the dispersed phase of the continuous liquid phase is listed as follows:

$$F_L = -C_L \varepsilon_G \rho_L (u_G - u_L) \times (\nabla \cdot u_L) \tag{7}$$

where $C_L$ is the horizontal lift coefficient.

Zhang [29] believed that in the fully developed area, the forces acting on the bubble mainly include the turbulent diffusion force and horizontal lift, and the horizontal lift coefficient $C_L$ and the turbulent diffusion force coefficient $C_{TD}$ are closely related according to the conservation of momentum. The specific expression is as follows:

$$\frac{C_L}{C_{TD}} = -0.2\frac{\varepsilon_L^2}{\varepsilon_L} \tag{8}$$

### 3.2.4. Wall Lubrication Force

Due to the effect of the wall, the liquid around the bubble is asymmetrical, so the bubble is subjected to a force away from the wall. This force is called wall lubrication force. Nguyen et al. [30] verified that liquid velocity relies on wall lubrication force. Therefore, the model of the wall lubrication force used in this part of the simulation was Tomiyama's equation [31]

$$F_{WL} = C_{WL}\rho_L \varepsilon_G |(u_L - u_G)|^2 n_W \tag{9}$$

The specific expression of $C_{WL}$ is:

$$C_{WL} = C_w \frac{d_b}{2}\left(\frac{1}{y_w^2} + \frac{1}{(D - y_w)^2}\right) \tag{10}$$

The definition of $C_W$ is:

$$C_w = \begin{cases} 0.47 & Eo < 1 \\ e^{-0.933Eo+0.179} & 1 \leq Eo \leq 5 \\ 0.00599Eo - 0.0187 & 5 < Eo \leq 33 \\ 0.179 & 33 \leq Eo \end{cases} \tag{11}$$

The expression of $E_O$ is:

$$Eo = \frac{g(\rho_L - \rho_G)d_B^2}{\sigma} \tag{12}$$

### 3.3. Bubble Breakup Model

Common bubble breakup models are: the Luo model [32], the Lehr model [33], the Ghadiri model, and the Laakkonen model [34]. However, the Luo bubble breakup model has the advantages of simple form, high prediction accuracy and wide application. Thus, in this study, the Luo bubble breakup model was adopted. The specific expression of the Luo model is shown in Equation (13):

$$\Omega_{br}(V, V') = K \int_{\zeta_{min}}^{1} \frac{(1 + \zeta)^2}{\zeta^n} \exp(-b\zeta^m)d\zeta \tag{13}$$

where $K$, $n$, $m$, $\beta$, $b$ can be specifically expressed as:

$$\begin{aligned} K &= 0.9238\varepsilon^{1/3}d^{-2/3}\alpha \\ n &= 11/3, m = -11/3, \beta = 2.047 \\ b &= 12\left[f^{2/3} + (1 - f)^{2/3} - 1\right]\sigma\rho^{-1}\varepsilon^{-2/3}d^{-5/3}\beta^{-1} \end{aligned} \tag{14}$$

### 3.4. Bubble Coalescence Model

Common bubble coalescence models include the Luo model, the free molecular model, and the turbulent-model. The bubble coalescence rate model can be expressed as:

$$\Omega_{ag}(V_iV_j) = \omega(V_iV_j)P(V_iV_j) \tag{15}$$

The collision frequency between bubbles can be expressed as:

$$\omega(V_iV_j) = \frac{\pi}{4}(d_i + d_j)n_in_j\overline{u}_{ij} \tag{16}$$

Based on Luo's coalescence efficiency model, the modified coefficient $Ce$ was introduced into the bubble coalescence efficiency model. The modified bubble coalescence efficiency model is listed as follows [35]:

$$Ce = 0.319\ln(\rho/\rho_0) + 0.665 \tag{17}$$

## 4. Results and Discussion

### 4.1. Mesh Independence

The numerical simulation was carried out by using the FLUENT 15.0 software as the platform. The CFD-PBM coupling model was presented and used to evaluate the gas holdup of the large and

small bubbles influenced by the critical bubble diameter, the superficial gas velocity, pressure, surface tension and viscosity in a large-scale high pressure bubble column. In the numerical simulation, the two-dimensional axisymmetric model could optimize the time of the calculation due to the small number of meshes. Figure 3a is a two-dimensional axisymmetric geometric model taken by the numerical simulation. Figure 3b is a grid map taken from the bottom of the column at 2000 and 3200 mm. The detailed information of meshing is shown in Reference [35].

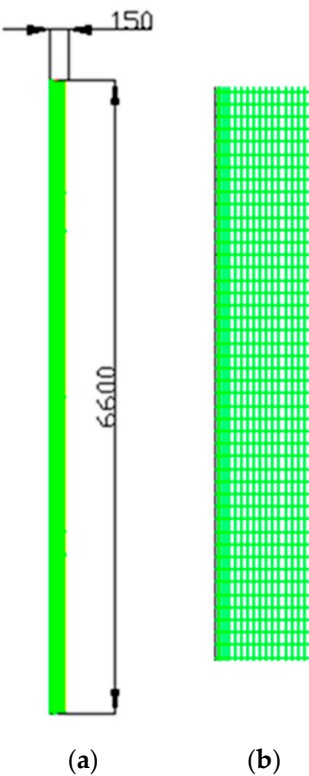

**Figure 3.** (**a**) Two-dimensional axisymmetric model. (**b**) A part of the meshing scheme.

The meshing of the geometric model had a great influence on the numerical simulation results. As the number of meshes increased, it not only requested improvements of the performance of the computer, it also increased the calculation time. Therefore, in order to improve the numerical simulation accuracy and the calculation efficiency, a suitable meshing method was the basis of the numerical simulation.

The mesh independence was investigated under the conditions of the superficial gas velocity of 0.317 m/s and a pressure of 0.5 MP, and, as such, four grids with grid numbers of 3960, 5940, 17160 and 47520 were used, respectively. The effects of meshing on radial gas holdup (Figure 4a), axial gas velocity (Figure 4b), and axial fluid velocity (Figure 4c) were verified. By comparison, it was found that selecting a grid with a grid number of 5940 could ensure the accuracy of calculation of each fluid mechanics parameter and could also satisfy a small calculation amount.

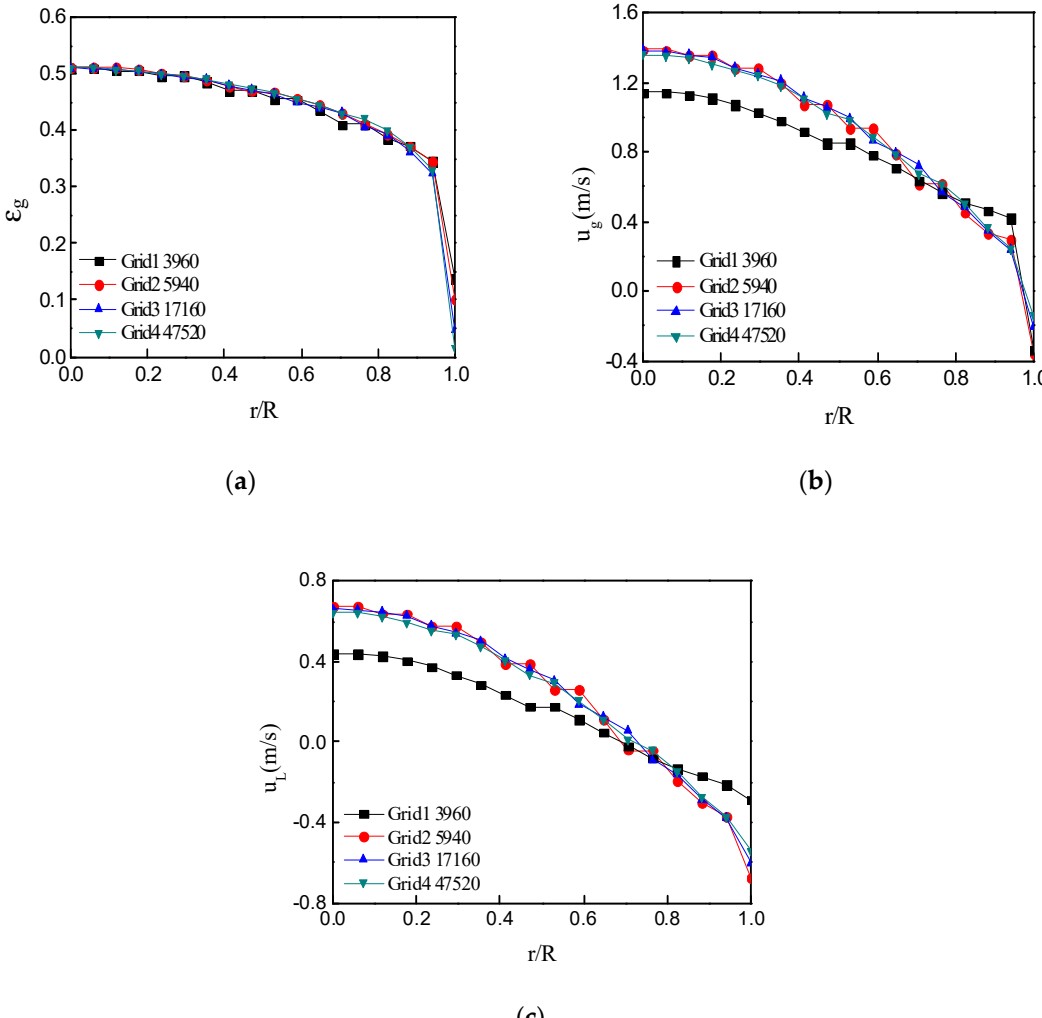

**Figure 4.** (**a**) Influence of grid partition on radial gas holdup. (**b**) Influence of grid partition on the axial gas velocity. (**c**) Influence of grid partition on the axial liquid velocity.

### 4.2. Determination of Critical Bubble Diameters

In the experiment, a bubble diameter of 3–6 mm could be regarded as small bubbles, and a bubble diameter of 10–80 mm could be regarded as large bubble [10–12]. Thus, it was very meaningful to distinguish bubble size from bubble swarm for calculating gas–liquid mass transfer characteristics. Xing et al. [20] used DGD to measure the tendency of the gas holdup of large bubbles with the superficial gas velocity in his experiment, and they gave a suggestive bubble critical value. In this study, we used 6, 7 and 8 mm as the critical bubble diameters to simulate the gas holdup, and we compared them with the experimental values obtained by DGD. From Figure 5, it can be seen that the results of simulation using different critical bubble diameters were different. It was found that the variation trend of the gas holdup of large bubbles was apparently consistent with the superficial gas velocity—that is, the gas holdup of large bubbles gradually increased with an increase of superficial gas velocity. However, when the critical bubble diameter was 6 mm, the simulation value of gas holdup of large bubbles was significantly higher than the experimental value. When the critical bubble diameter was 8 mm, the simulation value of gas holdup of the large bubble was lower than the experimental value. Therefore, the critical diameter of 7 mm of the bubble was adopted.

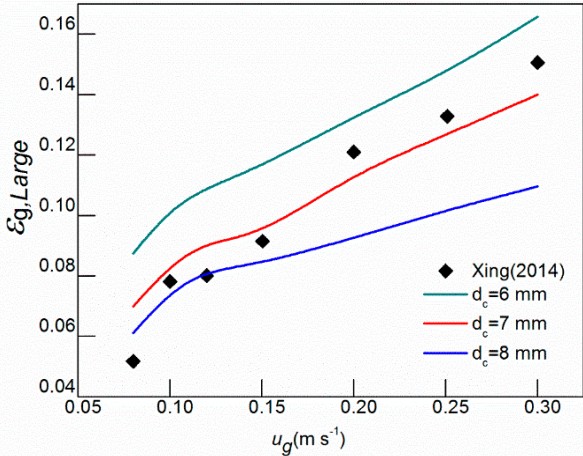

**Figure 5.** Effect of different critical bubble diameters on gas holdup of large bubbles.

### 4.3. The Gas Holdup of Large Bubbles and Small Bubbles

#### 4.3.1. Effect of the Superficial Gas Velocity on the Gas Holdup

Figure 6 shows the effect of superficial gas velocity on the average gas holdup, large bubbles gas holdup, and small bubbles gas holdup at different pressures (0.5, 1.0, 1.5, and 2.0 MPa). The effects of superficial gas velocity with different pressures on various gas holdup were simulated by the modified CFD-PBM coupling model. At the same time, compared the date obtained by Yang [19] using DGD, the variation of the gas holdup of large and small bubbles with the change of the superficial gas velocity was analyzed. It was found that the simulation value of the gas holdup of small bubbles was in good agreement with the experimental value. However, the gas holdup of large bubbles was slightly smaller than the experimental value. With the increase of superficial gas velocity, the gas holdup of large and small bubbles increased gradually, and the increased tendency of large bubbles was smaller than small bubbles. This variation trend is consistent with the experimental results of Xing [20]. The main reason for this phenomenon is that with the increase of superficial gas velocity, the turbulence within the column was intensified, and the bubble size was relatively uniform and smaller in diameter when the breakup and coalescence between bubbles were balanced. In the case that the critical bubble diameter was determined, the number of small bubbles increased and the number of large bubbles decreased. It can also be seen from Figure 6 that with the increase of pressure, the increased rate of the average gas holdup and the gas holdup of small bubbles gradually decreased with the increase of the superficial gas velocity. This is mainly because as the pressure got higher and higher, the bubble size got smaller and narrower. However, the increasing pressure had a slight influence on the bubble size. Under the determination of critical bubble diameter, the gas holdup of the small bubble was higher, but the small bubble gradually slowed down with the increase of superficial gas velocity.

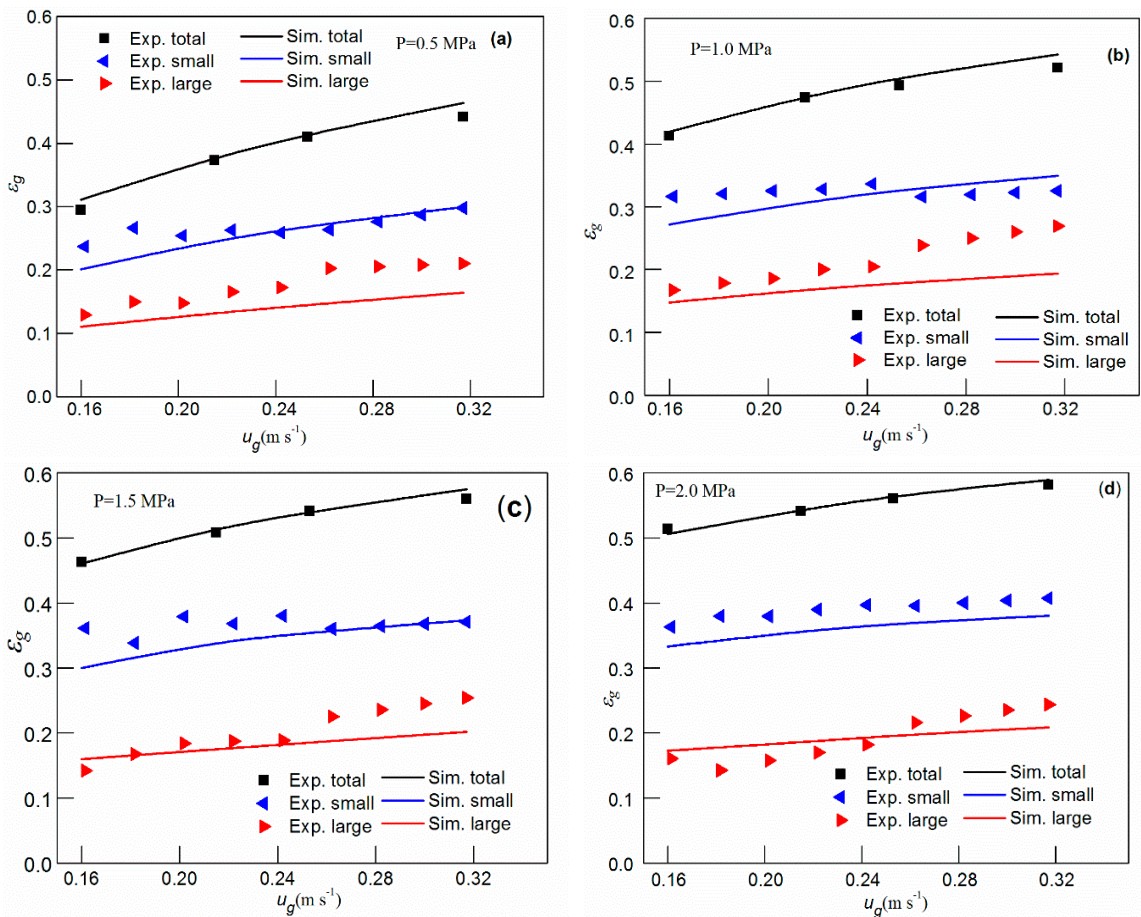

**Figure 6.** Effect of different superficial gas velocities on the gas holdup of large and small bubbles in different pressure ((**a**) P = 0.5 Mpa; (**b**) P = 1.0 Mpa; (**c**) P = 1.5 Mpa; (**d**) P = 2.0 Mpa).

4.3.2. Effect of the Different Pressure on the Gas Holdup

Figure 7 shows the effect of operating pressure on the gas holdup of large and small bubbles under different superficial gas velocities (0.160, 0.215, 0.253, and 0.317 m/s). The variation trend of the average gas holdup, large bubble gas holdup and small bubble gas holdup was investigated by numerical simulation. From Figure 7, it can be seen that the experimental values of Yang [19] and the values obtained through the CFD-PBM simulation are very consistent, especially the values of average gas holdup and small bubble gas holdup. For large bubble gas holdup, it can be seen that the simulated values were lower than the experimental values, especially when the pressure was higher, which makes the difference slightly obvious. This better illustrates that the modified CFD-PBM coupling model can predict the variation of gas holdup in the bubble column at different superficial gas velocities and different pressures. In addition, the gas holdup of large bubbles increased slowly with the increase of pressure. Moreover, compared with the gas holdup of large bubbles, the gas holdup of small bubbles increased significantly. The variation tendency of the gas holdup of large and small bubbles is consistent with the experimental results of Jordan et al. [36] and Krishna and Ellenberger et al. [10]. From Figure 7, it can be seen that when P ≤ 1 MPa, the average gas holdup and the gas holdup of large bubbles increased rapidly. When P ≥ 1 MPa, both increased slowly. Therefore, with the increase of pressure, the effect of pressure on the average gas holdup and the gas holdup of large bubbles gradually decreased.

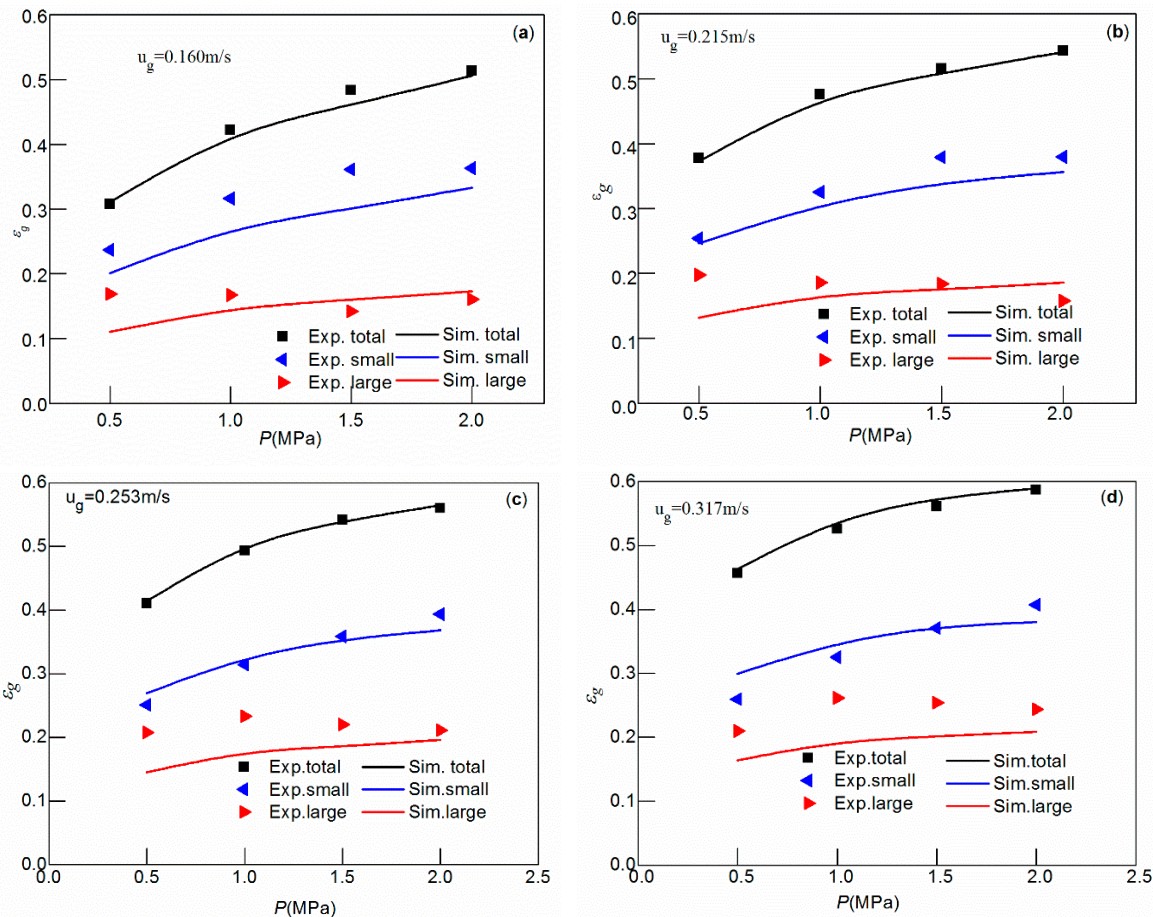

**Figure 7.** Effect of different pressure on the gas holdup of large bubbles and small bubbles at different superficial velocities ((**a**) $u_g$ = 0.160 m/s; (**b**) $u_g$ = 0.215 m/s; (**c**) $u_g$ = 0.253 m/s; (**d**) $u_g$ = 0.317 m/s).

### 4.3.3. Effect of Surface Tension on the Gas Holdup of Large and Small Bubbles

Figures 8 and 9 show the influence of different surface tensions ($\sigma$ = 49.9 × 10$^{-3}$, 60.7 × 10$^{-3}$, 66.7 × 10$^{-3}$, 70.0 × 10$^{-3}$ N/m) on the gas holdup of large and small bubbles at different superficial gas velocities under high-pressure conditions. From Figure 8, it can be seen that the gas holdup of small bubbles increased with the superficial gas velocity under different surface tensions. However, the gas holdup of small bubbles under a low surface tension was significantly higher than that under a high surface tension, which shows that the gas holdup of small bubbles gradually decreased with the increase of surface tension. Under the large surface tension ($\sigma$ = 70.0 × 10$^{-3}$ N/m), the simulated values agreed well with the experimental values. Under other surface tensions ($\sigma$ = 49.9 × 10$^{-3}$, 60.7 × 10$^{-3}$, 66.7 × 10$^{-3}$ N/m), the simulated value was consistent with the small bubble gas holdup measured by DGD, and there was a certain error. This may be because the critical bubble diameter was set too small. Thus, in the case of low surface tension, it helped to form small bubbles, and small bubbles rarely coalesced in the liquid phase. As such, the experimental gas holdup increased. From Figure 9, it can be seen that under different surface tensions, the gas holdup of large bubbles increased with the increase of superficial gas velocity, and the gas holdup of large bubbles increased with the increase of surface tension. The simulated value was in good agreement with the experimental value under a large surface tension. However, the simulated value of the large bubble under the small surface tension was larger than the experimental value, and, under the large surface tension, the experimental value was slightly higher than the simulated value.

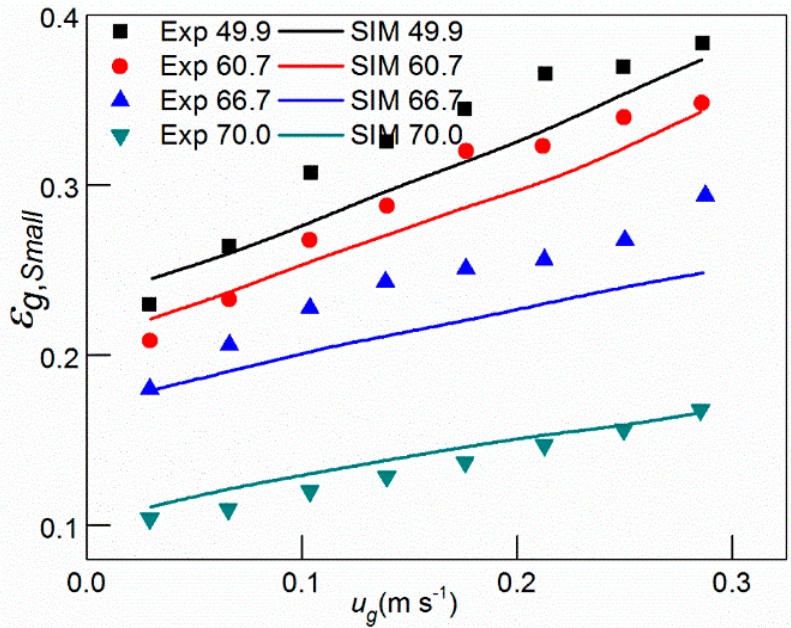

**Figure 8.** Effect of surface tension on the gas holdup of small bubbles.

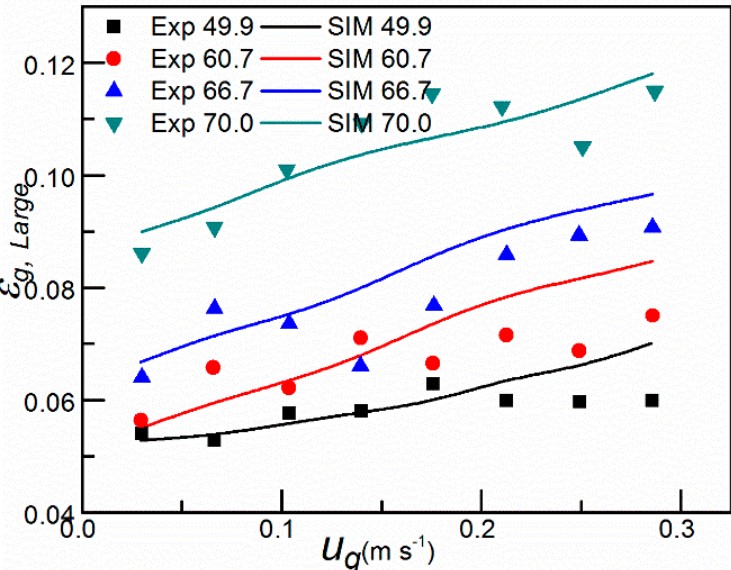

**Figure 9.** Effect of surface tension on the gas holdup of large bubbles.

By comparing the experimental and simulated values of the gas holdup of large and small bubbles, the modified CFD-PBM coupling model could basically investigate the influence of surface tension on the gas holdup of small bubbles in a high-pressure gas–liquid two-phase flow.

### 4.3.4. Effect of the Viscosity on the Gas Holdup

As can be seen from Figures 10 and 11, the modified CFD-PBM coupling model under high pressure was used to investigate the effect of different viscosities ($\mu = 1.41 \times 10^{-3}$, $1.91 \times 10^{-3}$, $2.35 \times 10^{-3}$, $3.54 \times 10^{-3}$ Pas) on the gas holdup of large and small bubbles.

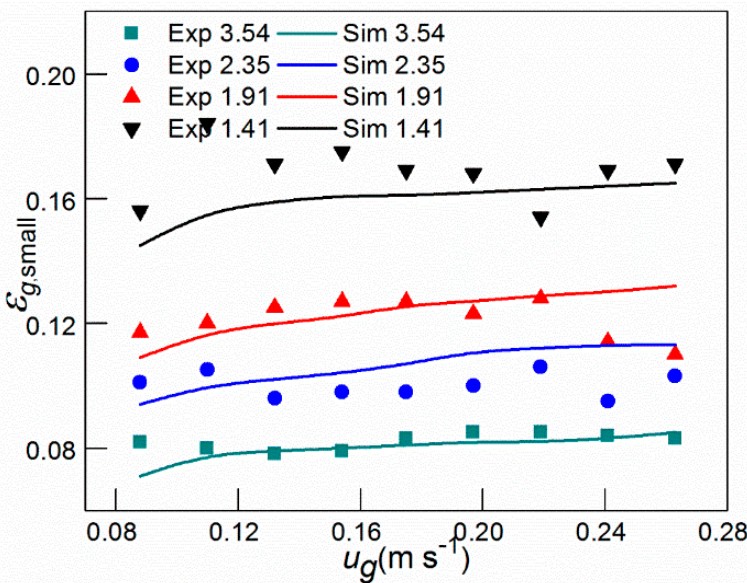

**Figure 10.** Effect of the viscosity on the gas holdup of small bubbles.

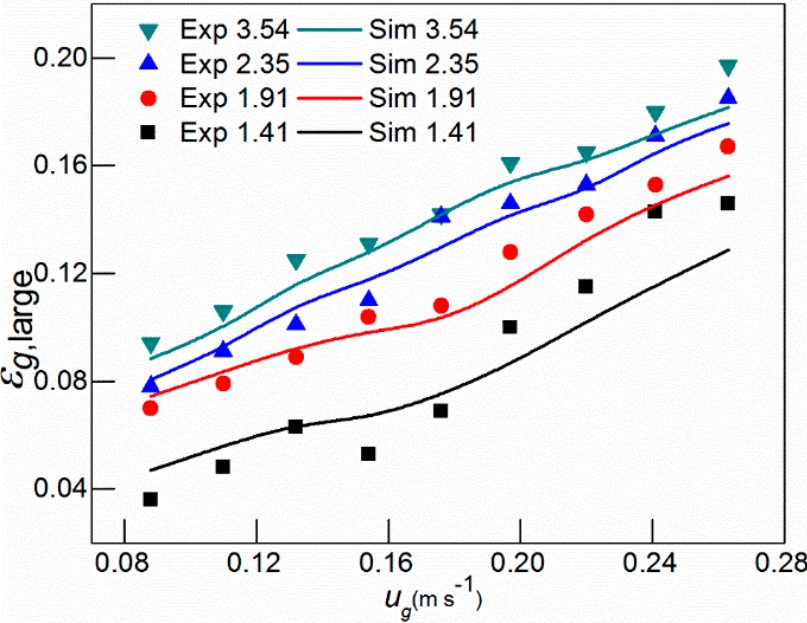

**Figure 11.** Effect of the viscosity on the gas holdup of large bubbles.

From Figures 10 and 11, it can be seen the gas holdup of small bubbles decreased with the increase of viscosity. With the increase of superficial gas velocity, the gas holdup first increased and then remained unchanged. The experimental results are consistent with results in Reference [20], mainly because the viscosity had little effect on the gas holdup in the case of low viscosity. As the viscosity gradually increased, the effect of viscosity on the gas holdup increased, resulting in a decrease of small bubbles in the column. The gas holdup of large bubbles increased with the increase of viscosity, and with the increase of superficial gas velocity, the gas holdup also increased. This was mainly because, with the increase of liquid viscosity, small bubbles in the column coalesced and formed large bubbles, which increased the bubble diameter and increased the gas holdup of large bubbles in the column. Yang [19] and Khare [37] also gave a reasonable explanation of the influence of viscosity on the gas holdup of large bubbles. The modified CFD-PBM coupling model was used to basically investigate the effect of viscosity on the gas holdup of small bubbles and to have a better prediction of the gas holdup in a high-pressure gas–liquid two-phase flow.

## 5. Conclusions

In this paper, the modified CFD-PBM coupling model by FLUENT 15.0 was used to simulate the high pressure gas–liquid two-phase flow in a bubble column, and the simulated values were compared with the experimental values. The main results were obtained as follows:

(1) Using 6, 7 and 8 mm as critical bubble diameters, the variation trend of the gas holdup of the large bubbles with the superficial gas velocity was obtained from the simulation results, and it was compared with the gas holdup of Xing [19] in the water–air system. It was finally determined the critical bubble diameter that divided the bubble into large and small bubbles was 7 mm.

(2) Using the modified CFD-PBM coupling model, the effects of superficial gas velocity and operating pressure on the gas holdup of large bubbles and small bubbles were analyzed. It is found that as the superficial gas velocity increased, the gas holdup of large and small bubbles increased to varying degrees. On the other hand, with the increase of pressure, the influence of pressure on the gas holdup of large bubbles gradually weakened. In the high pressure, the gas holdup of the small bubble increased with the increase of the superficial gas velocity.

(3) Compared with the results of the cold model experiment, it is found that the modified CFD-PBM coupling model could effectively estimate the influence of surface tension and viscosity on the gas holdup of large and small bubbles. That is, the gas holdup of the small bubbles gradually decreased as the surface tension and viscosity increased. The gas holdup of the large bubble gradually increased with the increase of the surface tension and viscosity.

**Author Contributions:** Conceptualization, F.T. and S.N.; methodology, B.Z.; software, B.Z.; validation, F.T., S.N. and B.Z.; formal analysis, F.T.; investigation, F.T.; resources, B.Z.; data curation, F.T.; writing—original draft preparation, S.N.; writing—review and editing, S.N.; supervision, H.J. and G.H.; project administration, H.J.

**Funding:** This research was funded by the National Natural Science Foundation of China, grant number 91634101 and The Project of Construction of Innovative Teams and Teacher Career Development for Universities and Colleges under Beijing Municipality, grant number IDHT20180508.

**Conflicts of Interest:** The authors declare no conflict of interest.

## Abbreviations

| | | |
|---|---|---|
| $\varepsilon_g$ | ——[–] | gas phase holdup |
| $dc$ | ——[mm] | The critical bubble diameter, mm |
| $u_i$ | ——[m s$^{-1}$] | velocity, m·s$^{-1}$, $i$ = 1: gas phase, $i$ = 2: liquid phase |
| $g$ | ——[m s$^{-2}$] | gravitational acceleration, m s$^{-2}$ |
| $\varepsilon_L$ | ——[–] | liquid phase holdup |
| $\rho$ | ——[kg m$^{-3}$] | density, kg·m$^{-3}$ |
| $U$ | ——[m s$^{-1}$] | velocity, m·s$^{-1}$ |
| $\tau$ | ——[–] | effective pressure tensor |
| $g$ | ——[m s$^2$] | gravitational acceleration, m·s$^2$ |
| $\varepsilon$ | ——[m$^2$ s$^{-3}$] | turbulent dissipation rate, m$^2$·s$^{-3}$ |
| $\mu_t$ | ——[Pa s] | turbulent viscosity, Pa·s |
| $K, k_L$ | ——[m$^2$ s$^{-2}$] | turbulent kinetic energy, m$^2$·s$^{-2}$ |
| $F_D$ | ——[N m$^{-3}$] | drag, N·m$^{-3}$ |
| $u_G$ | ——[m s$^{-1}$] | gas velocity, m·s$^{-1}$ |
| $u_L$ | ——[m s$^{-1}$] | liquid velocity, m·s$^{-1}$ |
| $C_D$ | ——[–] | drag coefficient |
| $C_{D,\infty}$ | ——[–] | ideal state drag coefficient |
| $Eo$ | ——[–] | parameter $Eo$ |
| $F_L$ | ——[N] | transverse lift, N |
| $C_L$ | ——[–] | transverse lift coefficient |

| | | | |
|---|---|---|---|
| $C_{TD}$ | ——[–] | | turbulent dispersion coefficient |
| $F_{TD,L}$ $F_{TD,G}$ | ——[N] | | turbulent dispersion force, N |
| $f_{TD,limiting}$ | ——[–] | | turbulent diffusion force model limiting function |
| $F_{WL}$ | ——[N] | | wall lubrication force, N |
| $C_{WL}$ | ——[–] | | wall lubrication coefficient |
| $Eo$ | ——[–] | | parameter $Eo$ |
| $d_B$ | ——[m] | | diameter of the bubble, m |
| $\Omega_{br}(V,V')$ | ——[–] | | bubble breakage rate |
| $\sigma$ | ——[N s$^{-1}$] | | surface tension, N·s$^{-1}$ |
| $\zeta$ | ——[–] | | relative diameter of the bubble |
| $\zeta_{min}$ | ——[–] | | minimum relative diameter of the bubble |
| $\Omega_{ag}(V_iV_j)$ | ——[–] | | bubble coalescence rate |
| $\omega(V_iV_j)$ | ——[m$^3$ s$^{-1}$] | | collision frequency between bubbles of size d$_i$ and d$_j$, m$^3$·s$^{-1}$ |
| $P(V_iV_j)$ | ——[–] | | bubble coalescence efficiency |
| $u_{ij}$ | ——[–] | | characteristic velocity of bubble collision |

**Lower subscript**

| | | |
|---|---|---|
| $G$ | —— | gas phase |
| L | —— | liquid phase |
| $i$ | —— | referring to the gas phase or the liquid phase |
| $b$ | —— | bubble |
| $i,j$ | —— | bubble section |

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
