# Peer review of "Simulation Study on Gas Holdup of Large and Small Bubbles in a High Pressure Gas–Liquid Bubble Column"

_processes, doi:10.3390/pr7090594_

Round 1

Reviewer 1 Report

In the Abstract:

"0.3m":Please add space between the number and the unit. It is recommended to use the same unit all over the text. “Large bubbles” and “Small bubbles”: Please write the critical values for large and small bubbles. This will make it easier for the reader to understand the rest of the abstract and the rest of the text. Confusing sentence: "With the increase of pressure, the gas holdup of small bubbles increases significantly, and the gas holdup of large bubbles almost change." Did you mean "the gas hold-up of large bubbles doesn't change? "the critical bubble diameter was analyzed and determined to
be 7 mm" 
What was the criteria used in the analyses? (write a short sentence explaining it in the Abstract, but the criteria for choosing the critical bubble diameter should also be well explained in the text.

In the Introduction:

"The flow structure in the bubble column was greatly...": Instead of "was", it should be "is". Cite references for the first two sentences of the second paragraph. What are "large" and "small" bubbles? Please explain these expressions in term of diameter. When you mention CFD-PBM, from which software do you mean (for instance, FLUENT, OpenFOAM)? Please explicitly write the software used when you mentioned CFD-PBM in the Introduction. Remove the last paragraph of the Introduction.

In the Experimental Set-up:

The legends and axes titles should have the same scale in both figures. In the caption, epsilon_g must be introduced (i.e., gas hold-up (epsilon_g)...)

In the subsection 3.1:

Very confusing: "In the Euler-Euler model treats, both water and bubbles are considered as the" : Please correct the sentence. Since in your case you have gas and liquid, the subscription "i" should be introduced as liquid or gas. Please substitute solid to gas. Please write the reference for the equations (if you took it from FLUENT Theory's Guide, mention it in the paper).

In the sub-subsection 3.2.1:

Why have you considered the correction proposed by Roghair et al.[18]? Please explain better this part. Please format the table in order to fit in one page, and add a space between the text and the table, and between the table and Eq.(3)

In the sub-subsection 3.2.3:

Please cite the reference for the first sentence ("When the bubbles move upward...")

In the subsection 3.4:

Cite the reference of Luo model and explain better about the efficiency of collision. Before introducing the modification proposed by Luo (Referenceneeded), the efficiency equation must be presented.

In the subsection 4.1:

Please, show a skecth with the line where the results for the mesh dependence study were taken. Figure 2 is not clear. Table 2: Does it show the mesh information of the selected mesh for the further simulations? (Please write a clearer caption for the table and remove 1 Cellzone, 5 face zones) A Richardson extrapolation is recommended for the mesh dependence study. Please add a table for this study (you can find more about this grid examination method in the book of Roache).

In the subsection 4.2:

Again, what do you mean by "large" bubbles? Please, at least once in the text, explain well when you mention "large" and "small" other times just put in a brackets the diameter range for each denomination.

In the sub-subsection 4.3.3:

Please correct the comma inside the brackets in the first sentence of the first paragraph (the same happens in the following subsection).

In the Conclusion:

Mention CFD-PBM from FLUENT-15.0

In general, the subject approached by the paper is very relevant and suits the scope of the journal. However, the writing must be improved (grammar mistakes corrected, and paragraphs written in a clear way) for the paper to be considered for publication.

The results and problem description must be discussed better, in order to provide for those reading the paper the relevance it contains.

Therefore, I suggest major modifications for the paper publication.

Thank you very much.

Reviewer 2 Report

The experimental method and the used material are ambiguous.
I can't understand some experssions in figures. 

(1)How did authors measured by the different pressure method?
If the details of the gas holdup measuring method have already been reported, please add these literatures.

(2)The conductivity probe method are probably point measuring method.
How many positions in plane did authors mesure? How did auhors procsss the data and convert to averaged values?

(3)page 4 Eq (1) solid phase ->gas phase
(4)Table1 Eorr ->Error??
(5)Fig. 4 (a) and Fig. 4 (b),Grid2 is not consistent.
Grid 1 3960
Grid2 5940
(6) In Table 2, LEVEL means the refiment LEVEL, doesnt it?
(7) DGD has two abbreviations that seem confusing.
That is diffrential pressure (line49 page2) and Dynamic Gas Disengagement(line57 page2).
The expression need to be changed.
(9) What was the range of E0 (Eotvos number ) value in Eq.(3) and (12)?
If possible, please add the range in text.
(10) P6 Please add the litertures about the Luo-model,Lehr-model,Ghadiri-model and LAakkonnen model.
(11) P7 Fig.3 (b), If possible, please add in figure the depth and width dimensions.
(12) In Eqs.(4)-(11), If the two phase simulation with ANSYS were performed with transient calculation, how did the data determine? Eventually,   which values  were used time average value or end time  value?
(13) The unit of key in figures 6-11 should be added, if possible.
(14) Fig.9:Subscript "Lar" ->"Large"
(15) Figures8 and 9,The surface tension was shown as a parameter in the Figure 8 and 9. What were mateial  used in experiment ?
(16) Identically, the viscosity  was shown as a parameter in the Figures 10 and 11.What were mateial  used in experiment?
(17)page 9, "in"  this study  -> (Capital letter) "In" this study

Round 2

Reviewer 1 Report

Dear authors,

Most of my comments have been taking into account in this new version (except the suggested for a the better mesh independence analysis suggested by Roache, P.J., Verification and Validation in Computational Science and Engineering, Hermosa Publishers, Albuquerque, New Mexico, 1998.). Although this suggestion was not included, due to the rest of the manuscript content, the paper is considered accepted from my side.

Thank you.
